# "If we work as a team, there are success stories." Unpacking team members' perceptions and experiences of what impacts team performance in a maternal and neonatal quality improvement programme in South Africa, before, and during COVID-19

Willem Odendaal [1,2,3]*, Ameena Goga[1,4], Mark Tomlinson[5,6], Yages Singh[1], Helen Schneider[7], Solange Mianda[7], Shuaib Kauchali[8], Carol Marshall[9], Terusha Chetty[1,10‡], Xanthe Hunt[5‡]

1 HIV and Other Infectious Diseases Research Unit, South African Medical Research Council, Cape Town, Western Cape, South Africa, 2 Department of Psychiatry, Stellenbosch University, Cape Town, Western Cape, South Africa, 3 Health Systems Research Unit, South African Medical Research Council, Cape Town, Western Cape, South Africa, 4 Department of Paediatrics and Child Health, University of Pretoria, Steve Biko Academic Hospital, Pretoria, Gauteng, South Africa, 5 Institute for Life Course Health Research, Stellenbosch University, Cape Town, Western Cape, South Africa, 6 School of Nursing and Midwifery, Queens University, Belfast, United Kingdom, 7 University of the Western Cape / South African Medical Research Council Health Services to Systems Research Unit, Cape Town, Western Cape, South Africa, 8 Division of Community Paediatrics, Department of Paediatrics and Child Health, University of the Witwatersrand, Johannesburg, Gauteng, South Africa, 9 South African National Department of Health, Pretoria, Gauteng, South Africa, 10 Discipline of Public Health Medicine, School of Nursing and Public Health, University of KwaZulu-Natal, Durban, KwaZulu-Natal, South Africa

‡ TC and XH are joint senior authors.
* willem.odendaal@mrc.ac.za

## Abstract

Many maternal and neonatal deaths and stillbirths can be avoided if quality of care is improved. The South African National Department of Health implemented a multi-partner quality improvement (QI) programme between 2018 and 2022, in 21 facilities, with the aim to reduce maternal and perinatal mortality. We conducted a qualitative evaluation to explore QI team members' perceptions of the factors shaping variation in team performance. The evaluation was conducted in 15 purposively selected facilities. We interviewed 47 team members from the 14 facilities consenting to participate in the evaluation, over three time points. We conducted 21 individual interviews and 18 group interviews. Data were thematically analysed using ATLAS.ti 8. Based on a preliminary assessment, six teams were rated as well-performing and eight, less well-performing. Patterns of divergence between well-performing and less well-performing teams were then examined through in-depth analysis of the full data set. Well-performing teams had a core team of members with a good understanding of the programme aims and QI methodology; a second in-charge member to ensure leader continuity; and leader stability throughout the implementation period. Well-performing teams were recruited from existing facility service teams who had a positive

**Data Availability Statement:** There are ethical restrictions on sharing a de-identified data set. Based on the geographic and demographic data provided in the paper, and due to the small sample size (participants and facilities) associated with this study, we are concerned that releasing the de-identified transcripts would put participant and facility confidentiality at risk. Many of the transcripts contain sensitive information about the internal workings of specific health facilities, which are tangential to the study, but would nonetheless have repercussions if made publicly available. Sara Cooper, a senior scientist at the South African Medical Research Council, and a non-author to our manuscript, can be contact if the dataset is sought and the first author is not able to respond to the request. She can be contacted at: sara. cooper@mrc.ac.za, Tel: (+)27-219380340.

**Funding:** The ELMA Philanthropies (20-P0001-C) (The ELMA Philanthropies to all evaluation fieldwork expenses), and the South African Medical Research Council, (South African Medical Research Council to WO and YS salaries) funded the evaluation. The funders had neither a role in the study design, data collection and analysis, nor in the decision to publish the study. The funders had no role in the preparation of this manuscript.

**Competing interests:** The authors have declared that no competing interests exist.

prevailing work culture. Team leaders' enthusiasm for QI and their ability to mobilise member buy-in, and how well teams worked together, further affected teams' performance. Existing facility contexts, how teams are set up, leadership—and member buy-in into the methodology, affect QI teams' performance. Focusing on these as well as supporting leaders to foster a shared vision and culture of excellence; mitigating contextual and implementation barriers; and strengthening team members' technical QI skills, has the potential to improve QI teams' performance.

## Introduction

Healthcare services are commonly provided by healthcare workers (HCWs) in teams [1]. It is for this reason that healthcare quality is often equated with effective teamwork [2], and not just with the skills of individual HCWs. Effective teamwork requires members to share their knowledge and expertise and have shared objectives, defined member roles, and effective communication [3, 4]. Conversely, teamwork barriers include high member turnover and lack of familiarity amongst members with each other, which often arises when members work different shifts [5]; and ineffective team leadership [6].

Moreover, teams are embedded within broader sets of contexts and processes that impact their performance [3, 5]. 'Broader contexts and processes' include alignment of team objectives with that of the organisation [3]; organisational culture and restructuring [6]; and (as reported by a neonatal resuscitation team), limitations imposed by the physical space they work in [4].

Quality improvement (QI) in healthcare includes amongst others, auditing service delivery, staff supervision, and strengthening supply chain management [7]. Relevant to this paper, is QI that comprises a HCW team addressing service delivery problems, with the aim of improving patients' health and experiences of care [8], and strengthening the professional practice and development of HCWs [9]. It comprises a collection of different models and techniques. Initially used in the manufacturing industry [10], QI gained traction in healthcare in the 1980's [11], and since then proliferated across all countries [12].

One of the most commonly used QI approaches is the Plan-Do-Study-Act (PDSA) model [13]. *Plan* entails developing small, low-cost solution/s (referred to as 'quality improvement plans' (QIPs)), to solve the service delivery problem [14]. The QIP is typically within the QI team's sphere of influence to effect change, and initially implemented on a small scale [15]. *Plan* also details how the team will assess the QIP's effectiveness. *Do* entails keeping record of implementing the QIP and assessing its effectiveness [16]. *Study* is when the team reviews the QIP outcomes, and in the *Act* phase that follows, the team decides to either abandon, adapt, or adopt the QIP into standard care [17]. Should the QIP initially fail, it can be amended and tested, again amended, or increased in scale, and iteratively assessed [18].

Many maternal and neonatal deaths, and stillbirths, can be prevented through improving quality of care. Evidence suggests that up to 60% of maternal deaths can be avoided by improved care [19]. Bhutta et al estimated that improved facility-level care can annually prevent up to 1.3 million neonatal deaths, approximately 531 000 stillbirths, and around 113 000 maternal deaths [20].

QI programme outcomes have however shown mixed results. In one study it improved community healthcare workers' visits during antenatal and postnatal care [21]; in another, increased the hours of Kangaroo Mother Care for preterm babies [22]; while also improving intrapartum fetal heart rate monitoring in a referral hospital during COVID-19 [23]. However,

in a systematic review, QI as a stand-alone intervention had no effect on patient health outcomes [9]. Two other reviews concluded that the methodological limitations of the studies precluded the unqualified endorsement of the PDSA model as an intervention to improve healthcare [12, 24]. Several factors have been hypothesised to explain the variation in QI outcomes, including team leadership [25]; facility and larger health system contexts [26]; methodological fidelity [13]; HCWs' motivation [27]; and teamwork [17].

In addition to the general teamwork factors described above, which shape team performance, there are other teamwork aspects pertinent to QI teams. These include that members who experience respect and trust within the team, are more likely to engage in improvement activities [28]. How well team members buy into the QI methodology [27], and their grasp of the QI methodology [11], further affect team performance. QI teams should also comprise the appropriate HCW cadres to be effective [29]. For example in a QIP aiming at keeping mother/infant pairs in care, having midwives, administrative staff, and peer supporters as team members, contributed to improving the retention in care [29].

Between 2018 and 2022, the South African National Department of Health (NDoH) developed a maternal and neonatal healthcare (MNH) QI programme called *Mphatlalatsane* (translated as 'the bright morning star'), to boost other national programmes to strengthen MNH services. *Mphatlalatsane* was a multi-partner programme, aiming to decrease maternal and neonatal mortality and stillbirths by up to 50% in 21 participating facilities [30]. The QI methodology for *Mphatlalatsane* was implemented through HCW QI teams who provided MNH services in participating facilities, with QI advisors providing technical support to the teams.

Our past analysis of the *Mphatlalatsane* QI teams [31, 32], points to the importance of the leader, teamwork, QI advisors, and other factors, in how these teams performed. However, these analyses focused on QI team leaders' and QI advisors' perceptions and experiences, and so reflect what leaders and advisors considered as important team attributes, and contextual and process factors, that shaped team performance. It is possible that team members have different insights about influencing factors. In this paper we fill this gap and foreground team members' voices on QI team performance. This paper also aims to contribute to the global literature on team members' experiences of contextual and implementation processes related to team performance.

## Materials and methods

Our process evaluation was nested in a broader, mixed-methods evaluation of *Mphatlalatsane*, that assessed effectiveness in reducing mortality rates, and improving maternal experiences of care and quality of care [33]. We received ethical approval from the South African Medical Research Council (SAMRC) in 2020 (EC019-11/2019), and Stellenbosch University, South Africa (S21/05/096).

### Setting

The 21 *Mphatlalatsane* facilities were spread across four health districts in three provinces: Mpumalanga, Limpopo, and the Eastern Cape (Fig 1). In each province, seven facilities were purposively selected to participate. Districts 1, 2 and 4 [34–36] are largely rural, District 3, mainly urbanised [34] (more detail about the districts can be found here [30]). The seven facilities reflected the referral pathways and were made up of: two primary healthcare (PHC) clinics, which refer patients mostly to two specific community health centres (CHCs), which then refer patients to two specific district hospitals. Patients requiring more advanced care, are then referred to one regional hospital.

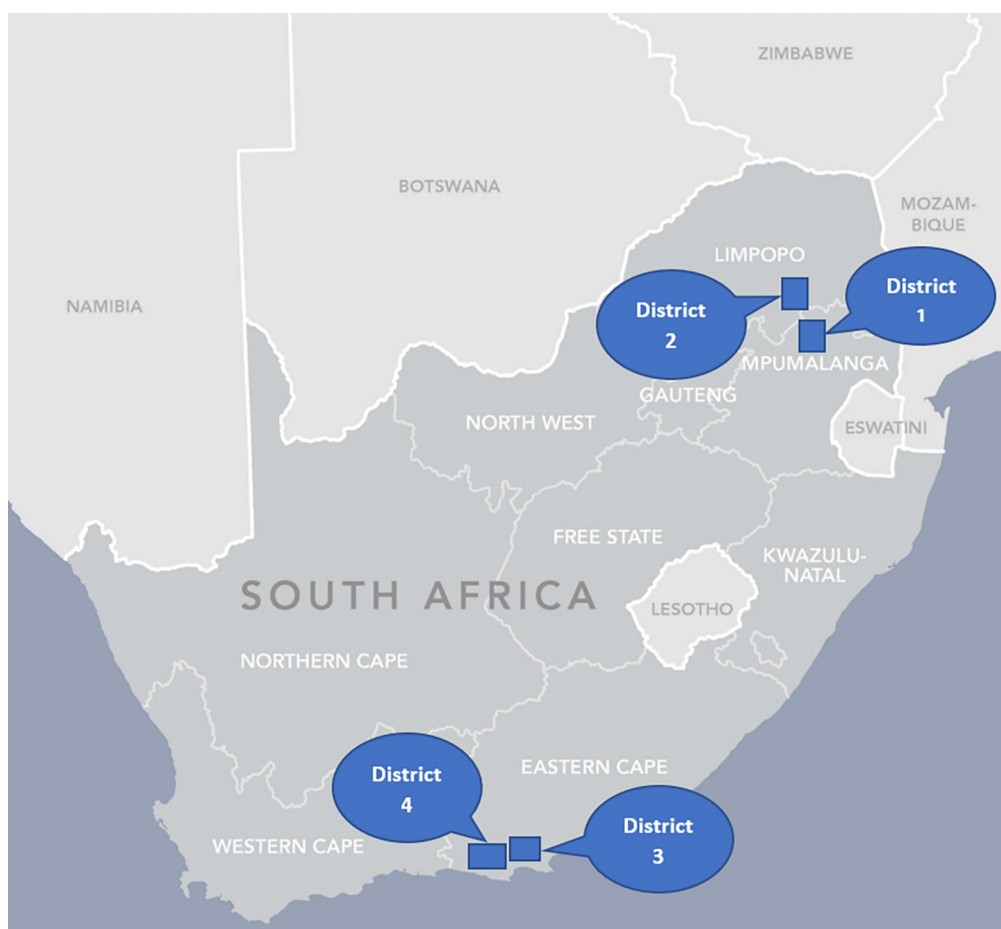

**Fig 1. Map of the *Mphatlalatsane* implementation districts.** Republished from SAMRC under a CC BY license, with permission from SAMRC Corporate and Marketing Communications Division, original copyright 2024.

The NDoH selected facilities with below-average to average performance on a set of indicators for institutional maternal mortality ratios. The Institute for Healthcare Improvement trained two to three HCWs, selected by facilities' management, on the QI approach, including the PDSA model. HCWs in Districts 2 (September 2019), and 3 and 4 (jointly in February 2020), received one training before the COVID-19 lockdown. District 1 HCWs had three trainings, one in September 2019 and two in March 2020. On return to their facilities, the HCWs and facility management recruited team members (henceforth referred to as 'members'), from existing staff. The facility management appointed one of the trained HCWs as team leader (henceforth referred to as 'leader'), and the teams had on average four to 12 members.

The selection of who to send for training and selecting the leaders and members, were purposively delegated to the management of each facility. This decentralised model was integral to the *Mphatlalatsane* design of fostering a grassroots level engagement and ownership. Consequently, the choice of the leaders, members, and team structure, in many instances mirrored the existing hierarchical structures in facilities.

The Clinton Health Access Initiative (CHAI), coordinated the *Mphatlalatsane* implementation and appointed QI advisors (henceforth referred to as 'advisors'), one each in Districts 1 and 2, with Districts 3 and 4 sharing the same advisor.

**Table 1. Participating facility type per district.**

| Type of facility | District 1 | District 2 | District 3 | District 4 | Total |
|---|---|---|---|---|---|
| Regional hospitals | 1 | 1* | - | - | **2** |
| District hospitals | 2 | 1 | - | 1 | **4** |
| Community healthcare centres | 2 | 2 | 1 | - | **5** |
| Primary healthcare clinics | 1 | 1 | - | 1 | **3** |
| **Total** | **6** | **5** | **1** | **2** | **14** |

*Facility with two teams

## Sampling

Due to budget constraints, we purposively selected 15 of the 21 *Mphatlalatsane* facilities for this evaluation. While all four facility types were included, we preferentially selected CHCs (n = 5) and hospitals (n = 6), that offered a larger range of MNH services than clinics (n = 3; Table 1). The facilities were distributed across urban and rural settings. Pre-*Mphatlalatsane*, the NDoH conducted a *Mphatlalatsane* readiness assessment in the 21 facilities, and our 15 facilities included facilities with a broad range of readiness assessment scores. Of the 15 facilities, one declined participation due to COVID-19 disruptions. Each facility had one team, but one had two, thus 15 teams from 14 facilities participated in this evaluation.

## Participants

A total of 47 members participated in the evaluation.

## Data collection

We conducted the evaluation between May 2021 and November 2022. Participant recruitment started on 28 April 2021 and ended 01 September 2022.

**Team member interviews.** We conducted a recruitment visit at each facility, during which we briefed leaders and requested they recruit members who had been in the team for more than six months. To include as many members as possible, we requested that they recruit different members for each data collection time point (Timepoint 1—May 2021, Timepoint 2—September 2021, and Endpoint—September 2022). Information documents, including the research aims and researcher's details and consent letters, were left with leaders to explain the evaluation to members they were going to recruit. At each timepoint, we briefed the recruited members and obtained their signed informed consent before their individual and group interviews. Though we requested individual interviews with members, we had member—group interviews (two to three members jointly), and in some instances, leaders choosing to join member—group interviews. There were 39 member data collection encounters: 21 member individual interviews, seven member—only group interviews, and 11 member plus leader—group interviews, with between two and three members and the leader, jointly (Table 2). Of the 47 members, six participated in more than one data encounter. There were on average four members per facility interviewed. The choice between individual interviews and group interviews were determined by the leader and members' availability, and on average 42 minutes long.

The interviews were face-to-face and took place in a facility-based office provided by the leaders. Interviews focused on members' views and experiences of participating in the QI teams, the leader's role, team performance, and implementation enablers and barriers. The interview schedule was piloted within the research team prior to the fieldwork. The interviews

**Table 2. Team member interviews.**

|  | District 1 | District 2 | District 3 | District 4 | Total |
|---|---|---|---|---|---|
|  | (6 facilities) | (5 facilities) | (1 facility) | (1 facility) |  |
| Member participants | 17 (3) | 22 (4) | 3 | 5 | **47 (4)** |
| Data collection encounters | | | | | |
| Member individual interview | 9 | 9 | 2 | 1 | **21** |
| Member—only group interviews | 3 | 3 | 0 | 1 | **7** |
| Member plus leader—group interviews | 3 | 5 | 2 | 1 | **11** |
| Total data collection encounters | **15** | **17** | **4** | **3** | **39** |

were conducted in English, audio recorded, and transcribed. WO (a male scientist at the SAMRC who holds a Masters (Research psychology), and with more than 18 years' qualitative research experience), collected all data.

**Fieldwork journal.** WO kept a fieldwork journal in which he recorded his fieldwork reflections.

## Analysis

During data collection, based on a preliminary examination of the emerging data, the analysis team (WO, XH, TC), classified the QI teams as well-performing, and less well-performing. This classification was a qualitative assessment of how well the team used the PDSA methodology over the implementation period, commencing with their 1st training to November 2022. For District 1 and 2 teams the period ran from September 2019 to November 2022, and for District 3 and 4 teams, from February 2020 to November 2022. Teams were rated as well-performing if they met all four criteria:

a. Presence of a leader and team structure, i.e., a group of HCWs who functions as the QI team;

b. Continuance of QI activities between August 2020 and March 2021 when the worst of the COVID-19 pandemic was over (we excluded the March to July 2020 period because NDoH had put all QI activities on hold);

c. Leaders were enthusiastic about QI and reported positively on their teams' performance; and

d. Leaders perceived their teams to be 'QI matured', i.e. the team could function without advisor support towards the final months of the implementation period.

The focus of the analysis was on understanding the drivers of difference between well- and less well-performing teams. This process involved an in-depth analysis of the full dataset. WO verified the transcripts against the recordings. The fieldwork journal and transcripts were analysed using ATLAS.ti, 8.1 (https://atlasti.com/). Applying Graneheim's and Lundman's [37] thematic analysis, WO coded five Timepoint 1 transcripts, discussed it with XH and TC, and amended the code list. After coding the remaining Time Point 1 transcripts, they again amended the code list. As new codes emerged from the next two timepoints' data, the analysis team amended the code list. We used the Consolidated Framework for Implementation Research (CFIR), with its emphasis on contexts and implementation processes, as framework for the analysis [38]. The codes were grouped into sub-themes, and from these, three overarching themes resulted: how team performance was shaped by (1) teamwork and the leader, and (2) team challenges; and (3) the issues that set well- and less well-performing teams apart from each other.

**Table 3. Team member nursing experience and years at the facility.**

|  | Median years in nursing (interquartile range) | Median years at the facility (interquartile range) |
|---|---|---|
| **District 1** | 27 | 22 |
| (6 facilities; 17 members) | (22–32) | (14–24) |
| **District 2** | 23 | 18 |
| (5 facilities; 22 members) | (20–26) | (11–20) |
| **District 3** | 32 | 18 |
| (1 facility; 3 members) | (31–32) | (13–22) |
| **District 4** | 8 | 3 |
| (1 facility; 5 members) | (7–9) | (2–3) |
| **Median** | 23 | 18 |
| (13 facilities; 47 members) | (20–30) | (11–23) |

## Ethics approval and consent to participate

We obtained ethical approval for the study from the South African Medical Research Council (EC019-11/2019ccc), and Stellenbosch University, South Africa (S21/05/096).

## Results

### Team member details

Most members (n = 46; 98%), were professional nurses and one a data clerk. District 4 members had a median of eight years nursing experience, and three years at the facility, versus 22 years nursing experience and a median of 17 years at their respective facilities for Districts 1–3 teams (Table 3). One District 4 leader opted not to have members participate in the evaluation, as she felt her participation sufficed for the purposes of the evaluation. Leaders and advisors were also interviewed, but their details are reported in another paper [32].

### Team performance assessment

Based on the preliminary data assessment, six of the 15 participating teams were classified as well-performing (Table 4): four in District 1, two in District 2, and none in Districts 3 and 4. Well-performing teams were predominantly at CHC and PHC level.

### Key thematic findings

We discuss the following three themes:

**Table 4. Well-performing teams by district and by facility type.**

| Facility type | District 1 | District 2 | District 3 | District 4 | Total |
|---|---|---|---|---|---|
| Regional hospitals | 1 of 1 | 0 of 1 | Declined participation | * | **1 of 2** |
| District hospitals | 0 of 2 | 0 of 1 | * | 0 of 1 | **0 of 4** |
| Community healthcare centres | 2 of 2 | 1 of 2 | 0 of 1 | * | **3 of 5** |
| Clinics | 1 of 1 | 1 of 1 | * | 0 of 1 | **2 of 3** |
| **Total** | **4 of 6** | **2 of 5** | **0 of 1** | **0 of 2** | **6 of 14** |

* No facility selected

1. Members' perceptions, firstly on teamwork and the leaders' role and skills, and secondly, on

2. Challenges, as factors that impacted team performance; and

3. Describe how teams' divergence on several issues, set the well- and less well-performing teams apart from each other. These issues were: the prevailing work culture in their existing facility service teams; how teams were set up; members' QI attributes; and their first QIP.

In the quotes, we notate members from well-performing teams as 'WP' and members from less well-performing teams, as 'LWP'. We also added the team number to the notation, e.g., 'WP 2' means it is a quote from team member of Well-performing team 2, and 'LWP 6' is a quote from a member of Less well-performing team 6.

**Theme 1: The importance of teamwork and leaders' roles and skills.** *Teamwork*. All aspects of the PDSA implementation, from identifying the service delivery problem, to developing, implementing, and assessing the QIP's effectiveness, required teamwork. For members, teamwork implied being given the opportunity to give input during the planning phases:

*So, solutions come from the other members and they say, okay, we can do this, we can do this. Then we choose a few that we see, have more ways [advantages] than the others. (WP 2)*

The importance of involving members in the planning of the QIPs, is shown in the following quote from a less well-performing facility. The QIP included a change in staff deployment, and the team member quoted here, expressed resistance to this because other members did not understand the logic behind the change:

*And then again on the other side [where the QIP was implemented], they didn't see the need why this sister [participated], because we placed this sister in maternity [there]. (LWP 6)*

Across well- and less well-performing teams, there was consensus that QI required HCWs to work together, and as expressed by this member, was the way to success:

*If we work as a team, there are success stories . . . So, I can tell them [those wanting to replicate Mphatlalatsane] that if they work as a team, it really helps, you'll see the results.. (WP 4)*

*Leaders' role and skills*. Without exception, members reported having good leaders, with views such as: *According to my assessment, she was a good leader. (LWP 1)*, and: *She's a leader and she is very excellent. She is hard working. She's available at any time that she's needed. (WP 1)*.

As illustrated in the quote from the less well-performing team, the member did not elaborate on why she said their leader was good, whilst the member of the well-performing was clear on why she thought their leader was good. A range of attributes were offered by members of well-performing teams as to why their leaders were good, which were mostly absent in the less well-performing teams. For some members of well-performing teams, it was because the leader was assertive and persistent in a positive way, in demanding quality work. Others found the leader's constant reminders why QI is important, inspired the team to participate in QI:

*As workers when they [leader] are introducing something, we say: "Oh, now they are introducing work to us." We don't always take it positively. But the way she was positive on this [QI] thing and the way she is, she reminds us about it. When there is a meeting she will influence us too until we love it. (WP 1)*

Other members praised their leaders for being approachable, which created a sense of being valued and fostered good team spirit:

*Yes, we also talk to our managers if we have any problems. They are approachable, we can approach them at any time, even if they are off, even at night. We call them any time that we've got problems. That makes us a team, a good team, this spirit. (WP 5)*

Leaders were also described as modelling commitment and "initiative":

*She [leader] is so committed. And also, initiative, because most of the time when we put the change ideas on the table . . . she's the one who initiated that and then you will just have to look at it and see whether it's suitable for us as a facility or not. But initiative and commitment, ja. (WP 3)*

This 'initiative' was often not limited to the QIPs, but included, for example, the leader who started a vegetable garden to ensure a better diet for TB patients. In another instance, the leader created a WhatsApp group for mothers to encourage long-acting family planning.

**Theme 2: Members' challenges.**    Three challenges affected QI activities, namely COVID-19, having members working night shifts, and staff shortages. COVID-19 caused serious emotional trauma, and most HCWs reported similarly as this member:

*. . . it was not easy for us, especially when it started that most of the staff were being COVID-19 positive . . . We were so very much afraid, let me be honest with you, but there was nothing that we're going to do. (WP 5)*

COVID-19 also severely disrupted service delivery and the QI activities: . . . *because of COVID, we stopped everything [QI] and we didn't continue. (LWP 7)*. Members reported how COVID-19 prevention measures, such as having only 50% of staff on duty, and not allowing face-to-face handovers at shift changing, impeded team meetings. Members on night shift found it challenging to participate in the QIPs, as was the case for this member who was only aware of the QIP but could not provide details about it:

*. . . last year October to May, I was working on night duty for those six months. Ja, what I know of, they were already in a process of taking a QIP . . . (LWP 1)*

Staff shortages were reported by leaders [32], and were confirmed by members: *Because of this shortage of staff . . . that is why [it is] going up-and-down with the QIPs, (LWP 1)*.

Members noted staff resources were not only about the HCW numbers, but also lacking sufficient senior staff to supervise juniors. This was more acute in the CHC and hospital teams than in the clinic teams. In the quote below, the member had advanced midwifery training, but in her maternity ward there were not enough on her level to supervise junior staff delivering routine services. Though this member was positive towards QI, she stated she did not have the capacity to participate in the QIP, in addition to overseeing the routine work of many juniors:

*But then the other problem is that there are juniors who cannot work on their own. (LWP 1)*

**Theme 3: Divergence between well- and less well-performing teams.**    *Prevailing culture in existing service teams*. QI team members were recruited from staff who were already organised in service team/s with specific duties, for instance if a midwife who provided ANC was

recruited into the QI team, she remained part of the ANC service team. QI members had thus 'dual membership', being member of an existing service team and of the QI team. Some members felt the prevailing work culture in their existing service team set the tone for their QI team, as illustrated by this member who felt motivated to improve the quality of care because of the example set by senior staff:

*I see the level of commitment from the [ward] managers [in the maternity unit]. They are committed, because if there's an adverse event, each and everybody is taking responsibility. I think that push us in the right direction. (WP 1)*

Members of less well-performing teams reported negatively about the culture in their existing service teams, as described by this member. She told us that whenever they (existing service team), were asking for extra hands to help with specific routine duties, staff declined:

*"No, I'm not trained!" They immediately say: "No, I'm not trained."* (LWP 6)

In another facility, several members complained about the negative atmosphere in their service team that made it unpleasant to work there:

*Like when you have the weekend off, you don't look forward to go to work on Monday because you don't know how it's going to be. (LWP 7)*

A member from another less well-performing team explained that in their facility it often happened that only the existing service team leader took responsibility for a programme, and that nothing happened in the leader's absence, because members did not take ownership of the programme:

*Then there were times, you know, people have got this ownership of programmes and if that person is not on duty, then the programmes relax. (LWP 5)*

On the opposite end were members from well-performing teams attesting to the positivity they experienced in their existing service teams, as shown in the contrasting experience of programme ownership in existing service teams. This member felt the prevailing norm of staff taking ownership when failing in a specific service delivery indicator, made it easier to work as a QI team:

*If the staff recognise the problem as their problem and not somebody else's problem, then it's easier [to implement QIPs]. (WP 2)*

Another member shared how their pre-*Mphatlalatsane* practice of meeting as an existing service team, was used to solve problems and support each other:

*Every morning, especially when we have identified something in the maternity section, maybe there was a pregnancy induced hypertension . . . we meet as professional nurses and discuss that case. And then we equip each other with the skills, how to manage preeclampsia . . . Yes, that's what we do. (LWP 3)*

*Team setup*. There was no systematic and uniform way in how the teams were set up as it was left to each facility. All well-performing teams had a core of 3–6 members taking charge of

planning and assessing the QIP, with larger ad hoc teams who implemented it. This setup was absent from the less well-performing teams, who in general had no members dedicated to QI planning and assessment. The core members were senior staff already managing others, and thus well placed to promote QI amongst ad hoc members. Long-term nursing experience and familiarity with the facility also boosted core members' impact on the QI team. This was evident in a facility where the core team members had between 34–37 years nursing experience and worked at the facility between 31–34 years. Here one of them described her role in the facility at large, and how it impacted their QI team:

*I was working in a Caesar ward, in a postnatal ward, so when she [colleague] is overloaded I used to go there and assist. So, we were just showing and acting as a role model. So that's why when we introduced this [QI] to our subordinates or colleagues, it was much easier because we were acting as a role model, so it was easy to copy. (WP 1)*

Leaders of well-performing teams also had a second in-charge, being one a core member, which participants noted as important, because leaders' routine work plus QI were often too much for the leader alone:

*Researcher: If you think about how you performed, was it because of her [leader]?*

*Member: I think it was because of her and the other sister [second in-charge] . . . Because that side, on the PHC side, there's too much paperwork and other meetings and the other things. So, if she [leader] was alone, it was not going to be excellent. (WP 3)*

A final difference between less well- and well-performing teams, was leader and member stability over time. Well-performing teams had the same leader all the time, and from what we observed, very stable membership. In six of the eight less well-performing teams, there were leadership changes. The negative impact of leader instability on team performance was magnified by the lack of a leader succession plan for these teams.

We found no evidence of functional core teams with senior members promoting QI in any less well-performing team. In four of them, members accounted how their 'core colleagues' left and were not replaced. Another typical situation in less well-performing teams was how members failed to generate interest from other staff members. They offered staff's workload, which was not different for well-performing teams, as reason for the failure:

*Most of the time it's me, and [the leader] and [another core member]. The others,*

they are not interested . . . Because sometimes we are busy in our units. (LWP 7)
*Members' QI attributes.* The first attribute was members' positive attitude towards QI, because it motivated staff (first quote), and yielded good results (second quote):

*They are more diligently done than before. Now, isn't it, we are looking at the goal that we're going to. Let's say that we're going to use our change ideas, they are smooth [no problems]. They are working more than before. (WP 5)*

*It [QI] is additional [burden], but at the end of the day they are good results. Like previously we were not taking the sputum from all pregnant women, but now when you are treated in ANC we take the sputum. (WP 4)*

This was contrasted with how members of less well-performing teams reported their attitude:

*She [the leader] wanted to show us how to do the statistics and the graph [QI activities], but we didn't have time, because I'm also doing circumcision that side. (LWP 5)*

The second attribute was members from well-performing teams had a good understanding of *Mphatlalatsane's* aims and QI methodology, for example understanding the importance of statistics to confirm QI outcomes (quote below). This contrasted with some members in less well-performing teams, who cited less robust evidence for QIP effectiveness, for example simply seeing more patients coming to seek services because of the QIP.

*Researcher: How do you do it [assess the QIP's effectiveness]?*

*Member: So, we do the auditing. We are busy on that every day. (WP 4)*

Their understanding of QI was also demonstrated in this member's recognition that QI is about using existing resources and changing what is within their control, both key tenets of the QI model.

*Researcher: Were you not sceptical about the additional workload that it's [QI] going to create?*

Member A: Oh, really, we were worried that this will need more staff, more equipment, what-what-what. But they said: "Guys, you must use what you have."

Member B: What's available.

*Member A: And, indeed, there was no additional staff, no additional equipment. But we managed. (WP 1)*

Members of less well-performing teams lacked this understanding, and some viewed *Mphatlalatsane* as equivalent to their QIPs, rather than a programme to reduce maternal/neonatal mortality and stillbirths, through improved care. In this quote, the team's QIP aimed to reduce unplanned pregnancy by educating all females of childbearing age visiting the facility, yet the member saw it as the *Mphatlalatsane* programme:

*Mphatlalatsane, I would describe it as a programme that caters for adolescents up until middle-aged women, which assists them . . . so that their pregnancy would be protected. (LWP 5)*

*A first, quick-win QIP.* The final issue setting teams apart, was differing responses to their first QIPs respectively from a well- and less well-performing team. In one instance, the former wanted to reduce complications during labour, while the latter aimed to improve partogram completeness. The well-performing team's 'Triage QIP' achieved their objective by identifying labour emergency cases quickly. The 'Partogram QIP' increased partogram completeness, but its health outcomes were not immediate, and it appeared as if these members did not view the prospective recording of medical indicators during labour, as potential triggers to medical interventions along the continuum of care after birth. Both QIPs required patient file auditing, yet those doing so for partogram completeness felt negative about the QIP, whilst the 'Triage QIP' team was excited about the immediately positive outcomes:

*Triage QIP*

We really saw that it was working for us: no women ever delivered outside because they are triaged in time and then taken to the delivery room for management. (WP 1)

Partogram QIP

*"So it was that they [members] complained because patients would be waiting and it would take time because they have to sit and audit. (LWP 1)*

## Discussion

We found that teamwork is essential when implementing QI, corroborating previous research [17]. As others found, leaders were important [39], and in our results, members of well- and less well-performing teams highlighted several the leaders' capabilities and qualities as factors important to QI team performance. This is also confirmed in our publication on the role of the *Mphatlalatsane* QI team leader in team performance [32], where we concluded: "Understanding the characteristics and skills of a team leader is crucial to contextualise a QI team's performance, and as such the outcomes of a QI programme." (p. 12). In that paper we recommend careful consideration of the leader selection criteria, for instance that leaders should show an aptitude for using routine data to improve the quality of care. Our study also highlights the importance of understanding the contexts and implementation processes in which QI teams function [27], as it impacts their performance.

We identified two issues that set well- and less well-performing teams apart, and thirdly, reflect on how COVID-19 and staff turn-over impacted team performance. It is important to note that though the team characteristics that strengthened performance were mostly absent in less well-performing teams, it will be an injustice to suggest that all of these were absent in all the less well-performing teams, all the time. As illustrated in the quote regarding a positive culture earlier, some less well-performing teams exhibited some characteristics at times, but in general performed less well because of inconsistent presence of facilitating characteristics.

### Prevailing work culture

The first was the prevailing work culture in the existing service teams. Members were recruited from existing service teams, yet, as is common, remained part of these existing service team [29]. This resulted in a double layered inner setting of existing—and QI teams. Well-performing teams were embedded in existing service teams with positive work-related norms and values. Their managers also modeled teamwork and programme ownership. This was contrasted with a culture of apathy towards work in the existing service teams for most of the less well-performing teams; in one instance described as a hostile environment. This confirms findings that what prevails within the 'micro system' of the existing service team, shape QI team performance [40].

### Team composition

Team set up and members' QI attributes relate to the prevailing culture. In the well-performing teams, senior HCWs were recruited into the QI team, becoming the core members who took responsibility for the QI work. They recruited ad-hoc members to implement the QIPs [41]. In *Mphatlalatsane*, the core members not only came from positive existing microsystems, but also displayed two QI specific attributes, namely buying into QI, and having a good understanding of *Mphatlalatsane's* aims and the QI methodology. Our results confirm previous research reporting that teams with influential members [40], who model improvement practices, and encourage other HWCs to participate [42], were better performing compared to teams without core members. Lacking the skills to conduct PDSA cycles are often part of why

teams fail [24], and therefore a sound understanding of the PDSA methodology, as found in well-performing teams, is important. It is reported that understanding programme goals motivates QI members to perform well [17, 42], and we conclude that because such understanding lacked in less well-performing teams, is part of why they performed less well. It is important that members' priorities for their patients align with the overall aim of the QI programme, as reported by the member who did not mind the additional work because QI benefitted the patients. A mismatch between patient and member priorities can become a performance barrier [43].

We did not find literature about the importance of a second in-charge member. In our view it was a good practice that ensured continuity in the leader's absence, given that some of the less well-performing teams did not function when leaders were absent. Although we assessed only one of the six hospitals as having a well-performing team, versus three of the five CHCs, and two of the three clinics, our data are insufficient to conclude that the facility type itself, shaped team performance.

The *Mphatlalatsane* management chose to leave the processes of selecting the leaders and members, and how to structure the teams, to each facility to manage. This clearly worked for the well-performing teams, and resulted in them having intrinsically motivated leaders and members who functioned in a team set-up that was conducive for their QI activities. The less well-performing teams may have benefited from a more structured approach and guidance on these matters.

## COVID-19 disruptions and staff turnover

Firstly, as experienced by the *Mphatlalatsane* QI teams, COVID-19 disrupted service delivery, with insufficient medical supplies or staff having to set up isolation wards [44, 45]. This was compounded by the psychological trauma HCWs suffered, including depression and burnout [46]. These disruptions and trauma were so severe that it put QI activities on hold for the first five months of the pandemic. However, HCWs' resilience [47], enabled well-performing teams to revive their QI work. Secondly, as reported elsewhere, staff shortages, resulting in work overload and less time for QI activities [43], and high member turnover [42], the latter found in the less well-performing teams, have been identified as barriers to team performance [26, 41]. We did not find literature on the third challenge to support our result that less well-performing teams had members working different shifts, which negatively affected their performance.

In reflecting on the intricacies of the issues that impacted QI teams' performance, we agree with Zamboni and colleagues that QI programmes are much more than tools and techniques, and closer to "social interventions" [26] (p. 15), than to narrow, clinical interventions with a specific methodology. Their performance is not only a function of how well they mastered the methodologies, but shaped by relationships and contexts.

## Recommendations

In the light of our results, we recommend the following to set up and manage QI teams for MNH in resource-constraint settings (Table 5).

## Strengths and limitations

A significant strength of this study is the gathering of data over three time points within the settings the QI teams functioned. This provided in-depth insight into their daily realities. These insights were augmented with the fieldwork journal data, that enabled nuanced data interpretations, illustrated in the following: the fieldwork notes that in many facilities there

**Table 5. Recommendations to recruit, train, and manage members for maternal and neonatal QI teams.**

|  | Recommendations |
|---|---|
| Recruiting and setting up | • At the start of the programme, the facility management should collectively discuss and agree on the selection criteria for the team leader and members.<br>• Recruit senior, experienced staff to form the "core team", who can have a positive influence over ad hoc team members.<br>• Identify a member who can act as second in-charge who has a good relationship with the leader.<br>• Ascertain how well the existing service team from which the members will be recruited, performs; if the service team is not performing well, address the issues before setting up the QI team. |
| Training | • Introduce QI as standard care and not as additional to existing service delivery guidelines.<br>• Balance training on QI techniques with teaching general teamwork principles.<br>• The second in-charge should attend all training the leader attends. |
| Managing | • Embed QI activities in existing practices and systems, for example use meetings of existing service teams for QI team meetings.<br>• Pay continued attention to members' enthusiasm towards their QI activities and address it when it declines.<br>• Solicit support for QI team activities and team stability from the senior management in the facility, highlighting the benefits for the performance of the wider facility service teams. |

were a lack of bed space for mothers, long patient queues, and staff shortages, made us understand why for some participants, QI was not a priority, given these dire working conditions. While the programme effectiveness on study outcomes remains to be seen, our evaluation will contribute to explaining these outcomes. This study may be limited by selection—and reporting bias as the leaders recruited member participants. While all members reported negatively on some areas of their teamwork, responses from well-performing teams were overwhelmingly positive in this regard. This may have been due to the recruitment of QI enthusiastic member or because leaders excluded participants with concerns. No members from less well-performing teams elaborated on why their leaders were not effective. Despite conducting the interviews in a private space, it was still within the facility where the leaders were busy with their daily routine. This, and when a leader joined member interviews, may have inhibited members from voicing constructive criticism of their leaders; however, 28 of the 39 data collection encounters were members-only, that reduced the risk of this occurring. Another participant bias limitation is the fact that we did not collect data on what the relationships between members in the group interviews were, and acknowledge that these may have impacted responses in some or the other way. Finally, we did not gather data on actual team sizes, and with the small member numbers in the evaluation, we cannot infer associations between the years of nursing experience or being based at a facility, and team performance. For the same reason we cannot infer an association between facility size and team size. Additionally, this study was not set up to investigate how team performance impact maternal and perinatal mortality.

## Conclusion

Establishing QI teams from existing service teams who have a positive work culture, is a key facilitator, or barrier, to how well a QI team will function. Investments to ensure that existing service teams work well together before establishing the QI team, is likely to pay dividends in the QI team's performance. QI team performance is not only a function of context, but also of implementation processes such as how teams are set up, and leader—and member stability. The importance of team leadership cannot be overemphasised. It is not training manuals that determine team performance, but the leader who moves beyond team management, fostering a shared vision and culture of excellence, is the one who unlocks performance. Continued

awareness of these variables and addressing barriers as they arise and / or strengthening enablers, will optimise QI teams' performance, and ultimately improve the health of mothers and their neonates.

## Supporting information

**S1 Table. Team member interview foci.**
(DOCX)

**S2 Table. Consolidated criteria for reporting qualitative studies (COREQ), 32-item checklist.**
(DOCX)

**S1 Fig. Code tree.**
(TIFF)

## Acknowledgments

Our sincerest appreciation to the five QI advisors for their enthusiastic participation. The data would have been much less rich without their insights and experiences. We are indebted to the HCWs who generously shared their time and insights with us. We also want to acknowledge the provincial and district management staff who supported the evaluation. The NDoH was the custodian of the *Mphatlalatsane* programme, and the Clinton Health Access Initiative acted as implementation secretariat. We thank Dr. Makua (NDoH) and Prof. Pillay (who was Director: Clinton Health Access Initiative, South Africa at the time of the study), who oversaw the intervention design and *Mphatlalatsane* implementation, for reviewing the manuscript.

## Author Contributions

**Conceptualization:** Willem Odendaal, Ameena Goga, Mark Tomlinson, Xanthe Hunt.

**Data curation:** Willem Odendaal.

**Formal analysis:** Willem Odendaal, Ameena Goga, Yages Singh, Terusha Chetty, Xanthe Hunt.

**Investigation:** Willem Odendaal.

**Methodology:** Willem Odendaal, Ameena Goga, Mark Tomlinson.

**Supervision:** Mark Tomlinson, Terusha Chetty, Xanthe Hunt.

**Writing – original draft:** Willem Odendaal.

**Writing – review & editing:** Willem Odendaal, Ameena Goga, Mark Tomlinson, Yages Singh, Helen Schneider, Solange Mianda, Shuaib Kauchali, Carol Marshall, Terusha Chetty, Xanthe Hunt.

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
