## [Decision Letter · Decision Letter 0]

16 Jan 2024

PGPH-D-23-01565

“If we work as a team, there are success stories.” Unpacking team members’ perceptions and experiences of what impacts team performance in a maternal and neonatal quality improvement programme in South Africa, before, and during COVID-19

Dear Dr. Odendaal,

Thank you for submitting your manuscript to PLOS Global Public Health. After careful consideration, we feel that it has merit but does not fully meet PLOS Global Public Health’s publication criteria as it currently stands. Therefore, we invite you to submit a revised version of the manuscript that addresses the points raised during the review process.

We look forward to receiving your revised manuscript.

Kind regards,

Kaveri Mayra

Academic Editor

Journal Requirements:

Additional Editor Comments (if provided):

Reviewers' comments:

Reviewer's Responses to Questions

**Comments to the Author**

1. Does this manuscript meet PLOS Global Public Health’s publication criteria? Is the manuscript technically sound, and do the data support the conclusions? The manuscript must describe methodologically and ethically rigorous research with conclusions that are appropriately drawn based on the data presented.

Reviewer #1: Yes

Reviewer #2: Yes

2. Has the statistical analysis been performed appropriately and rigorously?

Reviewer #1: N/A

Reviewer #2: N/A

3. Have the authors made all data underlying the findings in their manuscript fully available (please refer to the Data Availability Statement at the start of the manuscript PDF file)?

Reviewer #1: Yes

Reviewer #2: Yes

4. Is the manuscript presented in an intelligible fashion and written in standard English?

Reviewer #1: Yes

Reviewer #2: Yes

5. Review Comments to the Author

Reviewer #1: This is an important topic and addresses the complex interlinked issues at play affecting the implementation and sustainability of quality improvement initiatives, particularly for maternity services, which can lead to barriers to success for these important initiatives.

In the description of the intervention it states that management selected 2-3 HCWs for training in each facility. Facility managers then selected the team leader from these trained HCWs. It would be good to understand more about who the participants were, how they were selected, and how and on what basis team leaders were chosen, since composition of the teams is an important factor for WP teams. For example, was the selection based on seniority and experience? Was the selection process consistent across facilities?

Other team members mentioned include the second leader and the core team. How were these members selected and was this a designated part of the process of creating teams, or something that facilities themselves decided to do? Similarly, the first criteria for WP teams was the presence of a leader and team structure – can you clarify what is meant by team structure and how this was defined.

As noted in the limitations, having leaders present in some of the interviews would very likely have affected participants responses. Even having another team member in the interview could have affected participant’s responses if that person was the second leader or a member of the core team. A suggestion is to include in the brackets after each quote whether this quote is from an individual IDI, group interview or leader interview.

Leaders role and skills – under this heading it could be made more clear (if I am correct) that all teams, both WP and LWP, were positive about their leaders roles and skills, suggesting that this is not an issue that differs between WP and LWP teams. However, all the quotes in this section are from WP teams.

There is a major difference between the district 1-3 teams and the district 4 teams in terms of years of experience, which is highlighted in the text (L275). This suggests that the one facility in district 4 was very different from others. However, it is not clear, given that one of the teams in district 4 refused to participate, whether this one facility is the hospital or the clinic? In addition, there were only five participants in district 4 (3 interviews). Please clarify whether this is indeed an important difference.

The quote on L454 seems to suggest that LWP teams were also having a positive culture in their teams and there could be an additional comment to highlight this.

L328 both the quote and the sentence before it are unclear

The longitudinal data collection is an important strength of this study as noted in the discussion, but this is not being utilized in the narrative. You could include some quotes that demonstrate how some of the themes and sub-themes changed over the data collection period. A series of quotes from the same facility over time for example.

In L460 it is noted that all well-performing teams had 3-6 core members taking charge. It would be good to could give more explanation on how exactly how this was different in the LWP teams and how the size of the facility was related to this, given that most of the smaller facilities (CHCs and clinics) were WP.

In the conclusion the importance of team leadership is strongly highlighted as being key to success but the one thing that was common across WP and LWP teams was that they all praised their leaders.

Small corrections

The first sentence of the introduction could be revised to read better – just a suggestion

The sentence about abandoning, adapting or adopting (L95) needs rephrasing

Rephrase an average of (L171)

Typo L223 remove and at the end of the sentence

Reviewer #2: The study is about team members’ perspectives and experiences related to team performance in a quality improvement (QI) programme (Mphatlalatsane) implemented by the South African National Department of Health to reduce maternal/neonatal mortality and stillbirths, through improved quality care. The analysis provides a novel way of understanding the mixed results shown in other QI programmes by unpacking the differences between well and less-well performing teams. The usefulness of the study is that it identifies those factors associated with successful QI team performance so that these can be replicated elsewhere.

QI teams’ performance was affected by existing facility relationships and contexts, how teams are set up, member mastery of the methodology, and the ability of a team leaders to obtain the trust, rapport with and support of the team, which could mean going beyond the training manual to inspire and model for their team, motivation for the program. Less well performing teams lacked sufficient understanding of program goals, had fractured intra-personal dynamics that exacerbated existing challenges such as staff shortages, attitudes towards work, Covid-19 and other trauma.

While the teamwork aspects related to QI outcomes have been explored, the findings showcase specific institutional dynamics, for example the prevailing work culture, the team composition and the effects of Covid-19 on team performance. Table 5 offers immediately implementable recommendations for recruitment and set-up, training, and managing maternal and neonatal QIP teams in low-resource settings, which is a strength of the research. Given that the focus of the analysis was on understanding the drivers of difference between well- and less well-performing teams, it offers a road map to more successful QIP implementation.

A thorough analysis of the themes is offered by the authors who used a 32-item checklist for reporting on qualitative studies. This makes the findings robust with the data and analysis offered supporting the claims made. Although it is noted that the lead author kept a fieldwork journal, it is not clear to what extent this influenced the data analysis.

The implications of improved quality of care on health outcomes was not fully grasped by less-well performing teams. Where teams chose to focus on partogram completeness but neglected to ascertain the way this tool should be used to facilitate decision-making, trigger preparations for transfer, discussions with senior nurses and the need for medical action - then the quality improvement associated with the task was poorly understood. This point could be given more attention in the article as monitoring and evaluation of maternal and foetal indicators must occur prospectively so that indications of the need for medial interventions can be deciphered timeously. Instead it seemed to be occurring retrospectively, stopping nurses from attending to patients when it was necessary, and hence the QIP may even be negatively associated with worsening health outcomes where the link between quality and improved mortality and morbidity was not well understood. This is a point that should be raised in the article. Improvements in work processes are not always associated with improvements to patient care.

Line 575 "embedded in in existing" - repeated word

Line 599 "became a performance" - extra space

Line 223 "enablers and. The" - additional word

Line 400 "she did not having" - grammar

Line 444 "service teams This". missing full stop

6. PLOS authors have the option to publish the peer review history of their article (what does this mean?). If published, this will include your full peer review and any attached files.

**Do you want your identity to be public for this peer review?** For information about this choice, including consent withdrawal, please see our Privacy Policy.

Reviewer #1: No

Reviewer #2: **Yes: **Dr Nicole M Daniels

---

## [Decision Letter · Decision Letter 1]

10 Sep 2024

“If we work as a team, there are success stories.” Unpacking team members’ perceptions and experiences of what impacts team performance in a maternal and neonatal quality improvement programme in South Africa, before, and during COVID-19

PGPH-D-23-01565R1

Dear Mr Odendaal,

We are pleased to inform you that your manuscript '“If we work as a team, there are success stories.” Unpacking team members’ perceptions and experiences of what impacts team performance in a maternal and neonatal quality improvement programme in South Africa, before, and during COVID-19' has been provisionally accepted for publication in PLOS Global Public Health.

Best regards,

Julia Robinson

Executive Editor

Reviewer Comments (if any, and for reference):

Reviewer's Responses to Questions

**Comments to the Author**

1. If the authors have adequately addressed your comments raised in a previous round of review and you feel that this manuscript is now acceptable for publication, you may indicate that here to bypass the “Comments to the Author” section, enter your conflict of interest statement in the “Confidential to Editor” section, and submit your "Accept" recommendation.

Reviewer #2: All comments have been addressed

2. Does this manuscript meet PLOS Global Public Health’s publication criteria? Is the manuscript technically sound, and do the data support the conclusions? The manuscript must describe methodologically and ethically rigorous research with conclusions that are appropriately drawn based on the data presented.

Reviewer #2: Yes

3. Has the statistical analysis been performed appropriately and rigorously?

Reviewer #2: N/A

4. Have the authors made all data underlying the findings in their manuscript fully available (please refer to the Data Availability Statement at the start of the manuscript PDF file)?

Reviewer #2: Yes

5. Is the manuscript presented in an intelligible fashion and written in standard English?

Reviewer #2: Yes

6. Review Comments to the Author

Reviewer #2: Thank you for your thorough revisions and consideration of the comments.

7. PLOS authors have the option to publish the peer review history of their article (what does this mean?). If published, this will include your full peer review and any attached files.

**Do you want your identity to be public for this peer review?** For information about this choice, including consent withdrawal, please see our Privacy Policy.

Reviewer #2: **Yes: **Nicole M Daniels
